# Dexfenfluramine and Pergolide Cause Heart Valve Disease via Valve Metabolic Reprogramming and Ongoing Matrix Remodeling

**DOI:** 10.3390/ijms21114003

**Published:** 2020-06-03

**Authors:** Cécile Oury, Patrick Maréchal, Nathalie Donis, Alexia Hulin, Alexandre Hego, Julien Tridetti, Mai-Linh Nguyen, Raluca Dulgheru, Marianne Fillet, Alain Nchimi, Patrizio Lancellotti

**Affiliations:** 1Laboratory of Cardiology, GIGA Cardiovascular Sciences, University of Liège, 4000 Liège, Belgium; pmarechal@chuliege.be (P.M.); nathalie.donis@doct.uliege.be (N.D.); ahulin@uliege.be (A.H.); alexander.hego@uliege.be (A.H.); ju.tridetti@gmail.com (J.T.); mlnguyentrun@gmail.com (M.-L.N.); redulgheru@chuliege.be (R.D.); alainnchimi@gmail.com (A.N.); plancellotti@chuliege.be (P.L.); 2Laboratory for the Analysis of Medicines, Center for Interdisciplinary Research on Medicines (CIRM), Department of Pharmacy, University of Liège, 4000 Liège, Belgium; Marianne.Fillet@uliege.be; 3Gruppo Villa Maria Care and Research, Anthea Hospital, 70124 Bari, Italy

**Keywords:** serotonergic drugs, valvular heart disease, metabolism, rabbit model

## Abstract

Several clinical reports indicate that the use of amphetaminic anorectic drugs or ergot derivatives could cause valvular heart disease (VHD). We sought to investigate whether valvular lesions develop in response to long-term oral administration of these drugs and to identify drug-targeted biological processes that may lead to VHD. Treatment of New Zealand White rabbits with pergolide, dexfenfluramine, or high-dose serotonin for 16 weeks induced valvular alterations characterized by extracellular matrix remodeling. Transcriptome profiling of tricuspid valves using RNA sequencing revealed distinct patterns of differentially expressed genes (DEGs) that clustered according to the different treatments. Genes that were affected by the three treatments were functionally enriched for reduced cell metabolism processes. The two drugs yielded more changes in gene expression than serotonin and shared most of the DEGs. These DEGs were mostly enriched for decreased biosynthetic processes, increased cell-matrix interaction, and cell response to growth factors, including TGF-β, which was associated with p38 MAPK activation. Treatment with pergolide specifically affected genes involved in homeostasis, which was corroborated by the activation of the master regulator of cell energy homeostasis, AMPK-α, as well as decreased levels of metabolism-related miR-107. Thus, both pergolide and dexfenfluramine may cause VHD through valve metabolic reprogramming and matrix remodeling.

## 1. Introduction

Valvular heart diseases (VHD) have been associated with the use of amphetaminic anorectic drugs, including fenfluramine, benfluorex, and dexfenfluramine, and of ergot derivatives, such as those used for the treatment of Parkinson disease, particularly pergolide and cabergoline [1,2,3]. However, the mechanisms responsible for the development of drug-induced VHD have not been fully elucidated yet. Animal studies have pointed to serotonin and its 5-HT_2B_ receptor as a possible common pathogenic link between drug-induced VHD and carcinoid heart valve diseases, characterized by excess of plasma serotonin secreted by neuroendocrine tumors [4].

Using a rabbit model, we previously showed that long-term oral administration of high-dose serotonin causes VHD [5]. The tricuspid valve was affected with moderate to severe regurgitation in all serotonin-treated rabbits, while left-sided aortic and mitral valves were less frequently and severely involved. Histological examination indicated carcinoid-like plaques, thickened valves, and chondroid metaplasia, confirming that serotonin signaling could be involved in the development of VHD. However, whether this mechanism applies to the pathogenesis of other drug-induced VHD remains undetermined.

Few studies in rodents have depicted a role of 5-HT_2B_ receptors in valvular alterations induced by pergolide or amphetaminic anorectic drugs. The 5-HT_2B_ receptor antagonist, cyproheptadine, inhibited valvular toxicity triggered by long-term intraperitoneal injections of pergolide in rats [6]. Importantly, a recent study in mice indicated that valve remodeling induced by chronic infusion of nordexfenfluramine, the main metabolite of dexfenfluramine and benfluorex, occurred through a direct stimulation of 5-HT_2B_ receptors by this drug metabolite, independently of an increase in free serotonin [7]. Indeed, 5-HT_2B_ receptors blockage by antagonists or its deficiency in transgenic *Htr_2B_^-/-^* mice prevented nordexfenfluramine-induced valve lesions, whereas depletion of serotonin by tryptophan hydroxylase inhibitor *para*-chlorophenylalanine failed to do so.

Strikingly, it has been shown that drugs associated with VHD commonly display high affinity for 5-HT_2B_ receptors [8], while other drugs like phentermine or trazodone that lack such effects are either low-affinity agonists or antagonists of the same receptor. Thus, drug interaction with serotonin receptors might lead to VHD development through the activation of signaling pathways distinct from those triggered by the natural agonist serotonin. Notably, it has been reported that specific microscopic features can distinguish valve lesions induced by anorexigen exposure from those associated with rheumatic disease, floppy mitral valve syndrome, or carcinoid heart diseases [9].

We therefore hypothesized that different pathological mechanisms may underly the adverse valvular effects of amphetaminic anorectic drugs, ergot derivatives, and high-dose serotonin. Our study aimed at investigating the mechanisms of these drug-induced VHDs using a rabbit model of chronic oral administration of dexfenfluramine or pergolide, versus serotonin.

## 2. Results

### 2.1. Oral Serotonin, Pergolide, and Dexfenfluramine Cause Valvular Lesions with Distinct Histological Features

Sixteen-week administration of pergolide (0.5 mg/kg per day) or dexfenfluramine (10 mg/kg per day) in drinking water of adult female rabbits did not affect behavior or modify body weight as compared to non-treated or serotonin-treated (80 mg/kg per day) rabbits. The weight of explanted heart and ratio of heart to body weight remained unchanged. At the end of the procedure, pergolide and dexfenfluramine were detected in rabbit plasma, with concentrations equal to 0.074 ± 0.038 ng/mL and 2.08 ± 1.09 ng/mL, respectively.

Both treatments led to the development of histological alterations of aortic, mitral, and tricuspid valves (Figure 1). Serotonin (Figure 1B,U) and pergolide induced the formation of chondroid metaplasia in aortic hinge of all rabbits (Figure 1C,U) (*p* = 0.05), while it occurred in two rabbits out of three after dexfenfluramine treatment (Figure 1D,U) (*p* = 0.2). Pergolide led to aortic leaflet thickening (Figure 1V), with accumulation of glycosaminoglycan (GAG)-rich deposits in leaflets of the three valves (Figure 1G,K,O). GAG deposits were also observed in response to serotonin treatment (Figure 1F,J,N). The tricuspid valve (TV) leaflets of dexfenfluramine-treated rabbits were characterized by abundant disorganized collagen fibers and less GAG (Figure 1P,T). TV leaflets of pergolide- and dexfenfluramine-treated rabbits showed areas of connective tissue growth factor (CTGF) that were not observed in the serotonin-treated group (Appendix A). None of the treatments induced calcification or infiltration of macrophages (Appendix A) in valve leaflets. Thus, these histological observations indicate that oral administration of pergolide, dexfenfluramine, or serotonin all caused valvular lesions.

### 2.2. RNA Profiling Identifies Gene Expression Patterns of Drug-Induced Valvular Alterations

In order to further characterize differences of valvular alterations induced by the two drugs and serotonin, we used an unbiased RNA-sequencing approach to identify differentially regulated genes in tricuspid valves isolated from the three treatment groups compared to non-treated rabbits. 

Principal component analysis identified subgroups of individuals based on treatment (Figure 2A). Moreover, hierarchical clustering of differentially expressed genes (DEGs) identified four clusters of distinct expression patterns that corresponded to different treatments (Figure 2B). In terms of numbers of DEGs per treatment (including both protein-coding and non-coding RNA species), pergolide and dexfenfluramine modified the expression of 2692 and 3020 genes, respectively, whereas only 533 genes were altered by serotonin. As shown on the Venn diagram of Figure 2C, pergolide and dexfenfluramine shared a majority of DEGs (up to 78%), consistent with PCA and hierarchical clustering analysis, while only about 30% of genes (169 genes) modified in response to serotonin were common to drug treatments. Full lists of DEGs for all comparisons are provided in Appendix A. These first analyses indicated that the two drugs induced more similar RNA profile modifications than serotonin did.

RT-qPCR analyses were performed to validate our RNA-seq data (Figure 2D). Both techniques demonstrated the upregulation of *COL1A2*, and *MGP*, encoding collagen type I α2 chain and Matrix Gla protein, respectively, in response to the three treatments, upregulation of the ribosomal protein L11 gene *RPL11* by serotonin, upregulation of *HAPLN1* encoding Hyaluronan And Proteoglycan Link Protein 1 by dexfenfluramine, and downregulation of miR-107 in response to pergolide.

### 2.3. Gene Ontology Analyses Depict Enrichment of Distinct Biological Processes Leading to Valve Remodeling

To determine the biological functions associated with DEGs changes induced by the two drugs and serotonin, Gene Ontology (GO) analysis was performed. Lists of DEGs common to all treatments (165 genes), to the two drugs (1927 genes), and specific to each of the treatments (serotonin: 273 genes; pergolide: 572 genes; dexfenfluramine: 862 genes) were separately considered for the analysis (Figure 2C). Figure 3A visualized the biological processes (GO terms) enriched by the three treatments as well as some genes associated with one or more GO terms. The biological processes included cell respiration, glucose, fatty acid, and nucleotide metabolic processes, along with response to oxygen levels and muscle structure development. More specific trends in GO terms were shown in Figure 3B, indicating that the three treatments modified the expression of genes that mostly contributed to a decrease of these processes that were all related to cell metabolism. In accordance with this result, we observed an increase of phosphorylation of the central energy sensor AMP-activated protein kinase-α (AMPK-α) on residue Thr172, which reached statistical significance in response to serotonin and pergolide (*p* < 0.05) as compared to non-treated rabbit valves (Figure 3C).

The 1927 genes that were modified by pergolide and dexfenfluramine were enriched for decreased RNA processing, ribonucleoprotein complex biogenesis, and gene expression (Figure 4A,B). The two drugs also provoked an enrichment of genes involved in cell response to growth factors, including TGF-β, cell-matrix adhesion, collagen fibril organization, and blood vessel development. We thus determined whether treatments with the two drugs and serotonin would differentially trigger the activation of MAPK pathways. The p38 MAPK pathway is indeed activated downstream of TGF-β receptors and is required for TGF-β-induced myofibroblastic activation of valvular interstitial cells (VICs) [10]. While ERK phosphorylation was only induced by serotonin treatment (Figure 4C), we found significantly elevated p38 MAPK phosphorylation in valves of rabbits treated with pergolide that was also marginally induced by dexfenfluramine but not with serotonin.

We then searched for biological processes that would distinguish each treatment (Figure 5A–F). It appeared that, in sharp contrast to the two drugs together, serotonin caused an increase of ribonucleoprotein complex biogenesis, gene expression, and RNA processing (Figure 4A,B and Figure 5A,B). In agreement with the strong activation of AMPK-α, pergolide specifically affected genes promoting homeostatic process, and, to a lesser extent, organ growth (Figure 5C,D). Finally, dexfenfluramine treatment increased cytoskeletal organization and tissue morphogenesis (Figure 5E,F), which might corroborate the observed distinct changes induced on valve histology (Figure 1). Altogether, these results indicated that high-dose serotonin, pergolide, and dexfenfluramine cause valve remodeling through distinct molecular pathways and biological processes. 

## 3. Discussion

Our study is the first in vivo demonstration of adverse heart valve alterations induced by chronic oral administration of either an amphetaminic anorectic drug, dexfenfluramine, or an ergot derivative, pergolide. We uncovered a novel mechanism of drug-induced VHD relying on disruption of cell metabolism and associated extracellular matrix (ECM) remodeling.

A comparison of the effects of these drugs with those of a high dose of serotonin indicated that the mechanisms leading to valve lesions are likely to differ. Indeed, the three treatments yielded distinct valve histology modifications, signaling pathway activation, and differential changes in RNA profiles. Nevertheless, as proposed earlier [3,6,7], we could not exclude that the drugs might exert their effect by interacting with serotonin receptors. It is possible that the drugs might exhibit higher affinity for 5-HT_2B_ receptors than serotonin, thereby triggering differential cell responses [8]. The identity of the valve receptors mediating the drug effects remains to be investigated.

We also found that these treatments caused distinguished histological lesions with different ECM reorganization. Valve leaflets display a complex, multi-layered ECM structure that provides strength, flexibility, and resistance to bending and tensile stretch. ECM turnover is tightly regulated by VICs, the major cell type of the valves. Under pathological conditions, transition of VIC to an activated state enhances ECM synthesis and degradation [11,12]. Dysregulation of ECM turnover, i.e., ECM remodeling, is therefore a major contributor to leaflet dysfunction and VHD [13]. Our finding that dexfenfluramine led to distinct ECM remodeling as compared to serotonin is in agreement with a previous study on human pathological valves showing distinctive microscopic valve features after amphetaminic anorectic exposure compared to carcinoid heart diseases [9]. We observed that pergolide and serotonin resulted in higher GAG content, while dexfenfluramine led to accumulation of disorganized collagen fibers and less GAG. As observed in myxomatous mitral valve degeneration [14], high GAG content is known to contribute to leaflet weakening and valve prolapse, while collagen accumulation with less GAG is associated with leaflet stiffening as in rheumatic mitral stenosis.

In line with these observations, our RNA-seq data depicted that the two drugs, in contrast to serotonin, promoted cell responses to TGF-β and activated p38 MAPK. Notably, TGF-β-mediated p38 MAPK activation plays a key role in VIC transition to an activated state, and subsequent fibrosis [10,15]. Overall, our RNA profiling revealed differential changes in RNA expression patterns in response to the two drugs and serotonin. First, the number of DEGs was much higher upon drug treatment than with serotonin. Second, the two drugs shared up to 75% of DEGs, while only 5–6% of DEGs were shared with serotonin. More importantly, our data pointed out distinct biological processes triggered by the three treatments. Interestingly, the two drugs were found to slow down the valve biosynthetic process. In particular, pergolide promoted the homeostatic biological process.

Despite the recognized importance of valvular cell biology in the pathogenesis of VHD, there is a paucity of information regarding the metabolism of heart valves. However, the metabolic syndrome has long been associated with VHD such as calcific aortic stenosis, suggesting that cardiometabolic risk factors could actively participate in the pathophysiology of aortic valve disease [16]. Furthermore, it has been shown in mouse models that hyperglycemia and mild hyperlipidemia can induce aortic valve disease [17]. An in vitro study using porcine VICs provided a direct link between cell metabolism and cell-matrix interaction [18]. This study showed that an alteration in metabolic substrate availability could affect valvular phenotypic properties such as contractility, as shown in a collagen gel contraction assay. The authors proposed that persistent hyperglycemia could lead to valvulopathy through pathological collagen remodeling. Our study goes a step further by strongly suggesting that metabolism disturbance induced by serotonergic drugs like pergolide and dexfenfluramine may lead to valve disease through valve metabolic reprogramming and subsequent ECM remodeling. The fact that the two drugs led to a significant increase in cell-matrix adhesion further supports the existence of a link between cell metabolism and cell-mediated ECM remodeling [19]. As a general rule, energy and nutrient availability dictate the level and type of biosynthesis in the cell, including the synthesis of ECM, integrin-dependent cell–ECM interactions, and the amount of energy available to maintain cell viability and undergo cytoskeletal remodeling [20]. It is therefore noteworthy that genes modified by dexfenfluramine were enriched for cytoskeletal organization process. Furthermore, changes in cytoskeleton organization may promote VHD. For instance, RhoA signaling drives heart valve calcification [21], and mutations in *Flna* [22] and *Dchs1* [23] encoding a cadherin and a cytoskeleton protein involved in mechanotransduction, respectively, both lead to the development of myxomatous valve disease. Strikingly, VIC elongation that occurs under hypertensive conditions was associated with altered cell metabolism [24].

The impact of pergolide on valve metabolism was corroborated by a significant activation of the master regulator of cell energy homeostasis AMPK-α [25]. Once activated, AMPK triggers a switch from ATP-consuming anabolic pathways to ATP-generating catabolic pathways in order to preserve energy homeostasis. Biosynthetic processes, such as gluconeogenesis, glycogen synthesis, lipogenesis, cholesterol synthesis, and protein synthesis are inhibited, whereas glucose utilization, fatty acid oxidation, and mitochondrial biogenesis are stimulated. These pathways are dysregulated in obese patients presenting metabolic syndrome associated with diabetes, lipid abnormalities, and energy imbalances. In this context, AMPK has been an attractive drug target to manage lipid and glucose abnormalities and maintain energy homeostasis, alleviating cellular dysfunction in various cardiac diseases [26]. Our finding that pergolide, similarly to serotonin, activates AMPK is remarkable knowing that the role of this kinase in heart valve homeostasis was, until now, totally unknown.

In agreement with the effect of pergolide on valve metabolism, this drug also significantly reduced valvular miR-107 levels, a key regulator of the process of metabolic reprogramming [27], mainly controlling glycolysis. miR-107 was reported to be upregulated in obese mice, and its silencing improved glucose homeostasis and insulin sensitivity [27]. The role of miR-107 in VHD has never been investigated.

Few studies have analyzed valve transcriptomic profiles by RNA-seq and mostly focused on calcified bicuspid and tricuspid aortic valves [28,29]. These studies highlighted main active biological processes leading to abnormal leaflet architectural organization, alteration of ECM constituents, and valvular remodeling. For instance, the study by Guauque-Olarte et al. [28] showed an enrichment for genes involved in fibrosis in diseased aortic valves versus normal ones, thereby suggesting that matrix remodeling could represent an interesting therapeutic target. This is strikingly similar to our findings with the two drugs. Indeed, biological processes such as response to TGF-ß, regulation of cell death, and cellular response to growth factors that are affected by the two drugs all represent mechanisms leading to fibrosis. To our knowledge, no other studies investigated RNA expression profiles in diseased tricuspid valves nor drug-induced modifications of valvular RNA. Thus, our study shows for the first time that serotoninergic drugs disrupt the normal valve homeostatic process, through biological processes that have been associated with degenerative aortic valve calcification.

Taken together, our data indicate that long-term use of pergolide or dexfenfluramine causes VHD through distinct molecular mechanisms resulting in valve metabolic reprogramming (Figure 6). Besides activation of p38 MAPK signaling, pergolide targets AMPK- and miR-107-dependent homeostasis processes, and dexfenfluramine affects cytoskeletal organization. Drug-induced alterations of valve homeostasis and/or cytoskeleton may modify valvular cell–ECM interactions, leading to ECM remodeling and subsequent valvulopathy.

Further investigations should determine whether antagonists of 5-HT_2B_ receptors could reverse pergolide or dexfenfluramine effects or whether targeting the identified pathways could prevent drug-induced valve remodeling. The evaluation of valvular effects of serotonergic drugs could also be clinically relevant. More generally, our findings highlight a need for improving our understanding of the link between metabolism and valvular heart disease, with the aim to identify new treatments of these diseases, along with valvulo-metabolic risk factors, as part of the cardiometabolic risk assessment.

## 4. Materials and Methods

### 4.1. Rabbits

Thirty-two 8-week female New Zealand white rabbits (Crl:KBL) (2–2.5 Kg) were divided into 4 groups (n = 8) based on drug treatment: serotonin (80 mg/kg/day), pergolide (0.5 mg/kg/day) [6], dexfenfluramine (20 mg/kg/day) [7], and non-treated controls. Drugs were administered via drinking water. Body weight was determined weekly throughout the protocol. After 16 weeks, blood samples were taken via catheterized marginal ear vein under general anesthesia (intramuscular injection of droperidol (0.625 mg/kg), and of a mixture of xylasine (5 mg/kg), and ketamine (35 mg/kg)). Adequacy of anesthesia was monitored by palpebral and pedal withdrawal reflex. They were then euthanized by intra-cardiac injection of sodium pentobarbital (200 mg/kg) before collection of heart tissues. All rabbit experiments are approved by our institutional ethics review board (Ethics Committee of the University of Liege, Belgium) and conform to the guidelines from Directive 2010/63/EU of the European Parliament on the protection of animals used for scientific purposes.

### 4.2. Measurements of Pergolide and Dexfenfluramine in Rabbit Plasma

Dexfenfluramine and pergolide concentrations in rabbit serum were determined by liquid chromatography coupled with mass spectrometry (LC–Chip–MS/MS). Briefly, the chromatographic separation was achieved on a 1200 series LC-chip system (Agilent Technologies, Germany) using an Ultra High Capacity chip including a 500 nL trapping column and a 150 mm × 75 μm analytical column, both packed with a Zorbax 80SB 5 μm C_18_ phase (Agilent Technologies). The mobile phase was composed of H_2_O/FA (100:0.1, *v/v*) (A) and ACN/H_2_O/FA (90:10:0.1, *v/v/v*) (B) and used in gradient elution mode (from 15% to 69% B in 5 min). Mass spectrometric detection was performed using a 6340 Ion Trap equipped with a nanoelectrospray ionization source operating in positive mode (Agilent Technologies, Waldbronn) (transitions followed for dexfenfluramine: 232.0 −> 158.8; 186.8 m/z and for pergolide: 315.2 −> 207.9; 267.0 m/z). Finally, an Oasis µElution MCX 96-well plate (Waters, UK) was used to prepare the samples for the analysis. Fifty microliters of plasma was needed per experiment and all conditions were performed in duplicate and back-calculated using a calibration curve.

### 4.3. Histology and Immunohistochemistry

Rabbit hearts (n = 3 per treatment) were harvested after 16 weeks of treatment. Hearts were fixed in 4% paraformaldehyde, dehydrated through a graded ethanol series, cleared in xylene, and embedded in paraffin. Rabbit hearts were sectioned at 7 μm. Movat’s pentachrome staining (American MasterTech) and Picrosirius red staining (Sigma-aldrich) of tissue sections were performed according to manufacturer’s instructions. Heart valve leaflet area was measured from 4 to 10 different tissue sections and averaged. For RAM11 staining, slides were pretreated for 30 min using citric acid antigen retrieval in water bath at 96 °C. The primary antibodies used were RAM11 (Dako) and anti-rabbit CTGF (Abcam) antibodies. Slides were counterstained for 2min with hematoxylin-eosin. Images were captured using Olympus FSX100.

### 4.4. RNA Isolation

Tricuspid heart valves (n = 5 per treatment) were dissected and collected separately after 16 weeks of treatment. Valve tissues were snap-frozen in liquid nitrogen before being stored at −80 °C until RNA extraction. Total RNA was extracted using the miRNeasy Mini Kit (Qiagen).

### 4.5. RNA Sequencing, Data Processing, and Analysis

RNA integrity was verified on a Bioanalyser 2100 with RNA 6000 Nano chips (Agilent technologies, CA, USA). DV_200_ values, calculated from Bioanalyser traces, were comprised between 55% and 73% and did not differ between experimental groups. After ribosomal RNA depletion using the RiboGone^TM^ - Mammalian rRNA depletion kit (Clontech, CA, USA), libraries were prepared using the SMARTer® Stranded RNA-Seq Kit (Clontech, CA, USA) following manufacturer’s instructions. Libraries were validated on a Bioanalyser DNA 1000 chip and quantified by qPCR using the KAPA library quantification kit. Sequencing of multiplexed libraries was performed on an Illumina NextSeq500 sequencer generating a total of around 10 million mapped reads per library on average. Raw reads were first demultiplexed and adapter-trimmed using Illumina bcl2fastq conversion software v2.17; then, the first three nucleotides of each read (added during the SMARTer® protocol) were removed using the FATSX Toolkit (v0.0.13). After quality control with FastQC (http://www.bioinformatics.babraham.ac.uk/projects/fastqc/), reads were aligned to the rabbit oryCun2 reference genome using Tophat v2.1.1 [30]. Reads were first aligned to a virtual transcriptome generated based on the annotation gene models, and then reads that did not fully map were aligned to the genome, spliced as needed. Quality of the sequencing data was controlled using Picard tools (http://broadinstitute.github.io/picard/). For the differential expression analysis at the gene level, raw gene counts were generated using HTseq count in “union” mode [31].

Independent filtering was done to remove uninformative genes. The proposed filtering method was applied in the *HTSFilter* package within the Bioconductor package in R (limma: Linear Models for Microarray and RNA-Seq Data User’s Guide) [32]. Prior to differential analysis, unwanted variations were identified using the *svaseq ()* and *RUVr ()* functions in R Bioconductor package *RUVSeq* [33]. From 23,669 detectable ENSEMBL genes, differentially expressed genes (DEGs) between treatment groups and controls were identified using limma-voom method from the Bioconductor/R package limma, using empirical Bayes statistical methods. A gene is considered differentially expressed if its corresponding FDR is less than 5%. Multiple comparison was handled by Benjamini–Hochberg approach. The RNAseq data have been deposited in NCBI’s Gene Expression Omnibus and are accessible through GEO Series accession number GSE113082 (https://www.ncbi.nlm.nih.gov/geo/query/acc.cgi?acc = GSE113082).

### 4.6. Gene Enrichment Analysis

Common or specific differentially expressed genes with an FDR < 0.05 were functionally annotated by using Database for Annotation, Visualization, and Integrated Discovery (DAVID version 6.8), and filtered out with database GOTERM_BP_FAT. Gene ontology (GO) Biological Processes with a Benjamini–Hochberg FDR-adjusted *p*-value < 0.05 were selected. Visualization of GO terms was performed using the GOPlot R package and GOCircle and GOChord function.

### 4.7. Real-Time Quantitative PCR

Total RNA extracted from rabbit tricuspid valves was reverse transcribed into cDNA using RevertAid H Minus First Strand cDNA Synthesis kit (Thermo Fisher Scientific) according to manufacturer’s instructions. Quantitative real-time PCR analyses were performed using SYBR PCR mix (Takara). Primer sequences can be found in Appendix A. Changes in gene expression were calculated by *ΔΔ*C_t_ method using *SNORA63* for normalization. The average of control samples was then set to 1 for each gene.

### 4.8. Protein Extraction and Western Blotting

Proteins were isolated from the organic phenol phase of QIAzol reagent-treated valvular tissues according to manufacturer’s protocol (Qiagen). Western blotting was performed according to classical procedures using the following primary antibodies purchased from Cell Signalling: phospho-p38 MAPK (Thr180-Tyr182), p38, phospho-p44/42 MAPK (Erk1/2) (Thr202/Tyr204), Erk1/2, phospho-AMPKα (Thr172), AMPKα. Blots were analyzed with ImageStudio software (LI-COR).

### 4.9. Statistics

Data normality was assessed using the quantile–quantile plot and histogram of error terms. Departure from normality was handled either by log transformation or non-parametric approach. Multiple comparisons between treated rabbits and controls were done using one-way ANOVA or Kruskal–Wallis test followed by Dunnett’s post hoc test with control sample as the reference. Comparison between two groups was performed using Mann–Whitney test. The appropriate statistical test is mentioned in the figure legend. All statistical analyses were performed using SAS 9.4 (SAS Institute Inc., Cary, NC, USA). Data are reported as dot plots, where each dot represents a rabbit and the horizontal bar indicates the means ± SEM using PRISM5 software package (GraphPad). A *p*-value < 0.05 was considered significant.

## Figures and Tables

**Figure 1 ijms-21-04003-f001:**
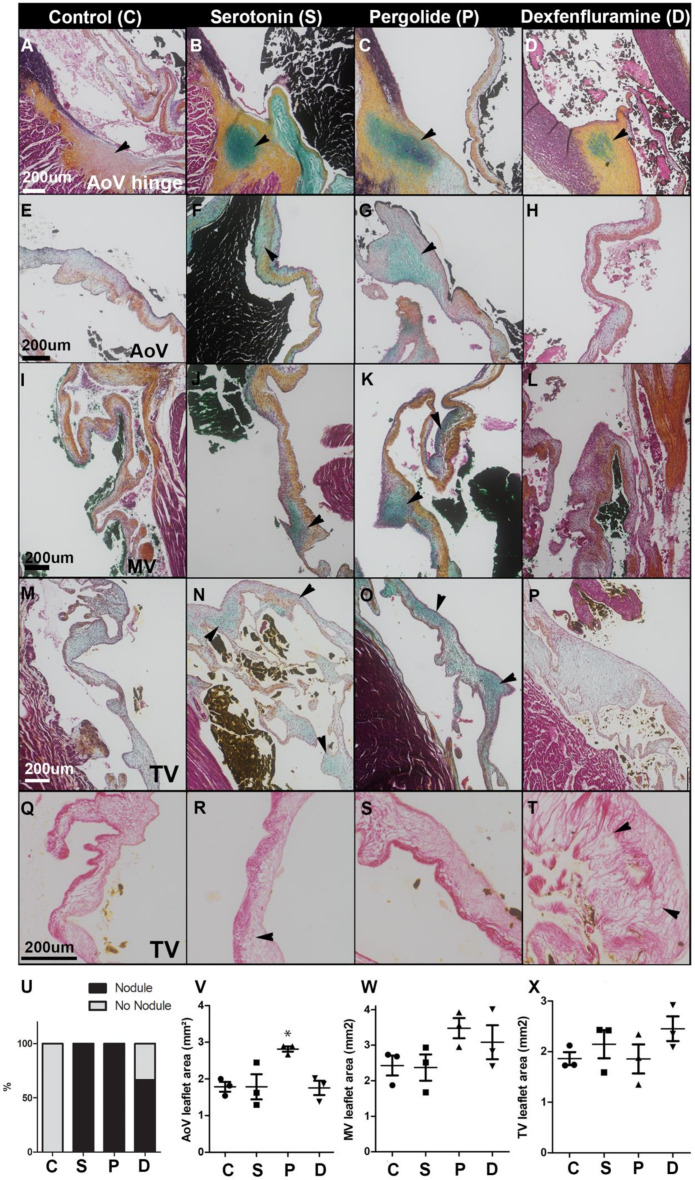
Drug treatments induce distinct valvular lesions: (**A**–**D**): Representative Movat’s pentachrome staining performed on heart section from control (**A**) and serotonin- (**B**), pergolide- (**C**), and dexfenfluramine- (**D**) treated rabbits. Arrowheads indicate chondroid metaplasia. (**E**–**P**): Representative Movat’s pentachrome staining performed on aortic valve (AoV) (**E-H**), mitral valve (MV) (**I**–**L**), and tricuspid valve (TV) (**M**–**P**) leaflet sections from control (**E**,**I**,**M**), serotonin (**F**,**J**,**N**), pergolide (**G**,**K**,**O**), and dexfenfluramine (**H**,**L**,**P**). Arrowheads indicate GAG-rich deposit. (**Q**–**T**): Picrosirius red staining performed on TV leaflet section from control (**Q**) and serotonin- (**R**), pergolide- (**S**), and dexfenfluramine- (**T**) treated rabbits. Arrowhead indicates remodeled collagen layer. (**U**): Percentage of hearts displaying chondroid metaplasia at the aortic valve hinge. (**V**–**X**): Leaflet area was measured and averaged from at least 4 different sections per valve and rabbit (n = 3). Data are reported as dot plots where each dot represents a rabbit and the horizontal bar indicates the mean ±SEM. * *p* < 0.05. Kruskal–Wallis test was used to determine statistical significance (n = 3). C = control, S = Serotonin, P = Pergolide, and D = Dexfenfluramine. Scale bars = 200 μm.

**Figure 2 ijms-21-04003-f002:**
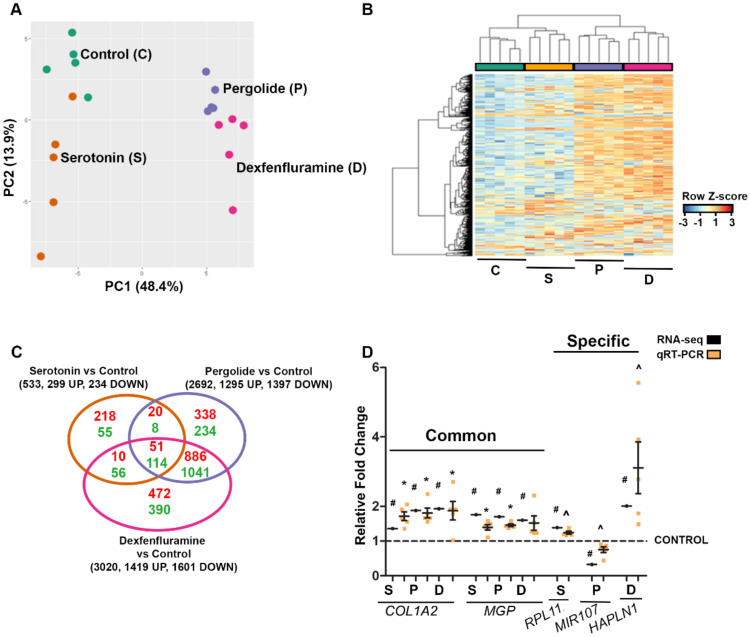
RNA-seq analysis of tricuspid valve (TV) transcriptomes indicate that Pergolide and Dexfenfluramine treatment shares the majority of differentially expressed genes (DEGs). (**A**): Principal component analysis (PCA) on tricuspid valve leaflets treated with water (green dots, n = 5), serotonin (orange dots, n = 5), pergolide (purple dots, n = 5), and dexfenfluramine (pink dots, n = 5). (**B**): Heatmap view of z-scored expression profile of DEGs in TV of rabbits treated with serotonin (n = 5), pergolide (n = 5), and dexfenfluramine (n = 5) compared to control (n = 5). (**C**): Venn diagram displays comparison between DEGs for each drug treatment. (**D**): Data of qRT-PCR (n = 5 per treatment) are reported as dot plots where each dot represents a rabbit and the horizontal bar indicates the mean ±SEM as compared to relative fold change obtained by RNA-seq analysis. # FDR <0.05, * *p* < 0.05 determined with one-way ANOVA followed by Dunnett’s multiple comparison test (n = 5 for each treatment). ^ <0.05 determined with Mann–Whitney test.

**Figure 3 ijms-21-04003-f003:**
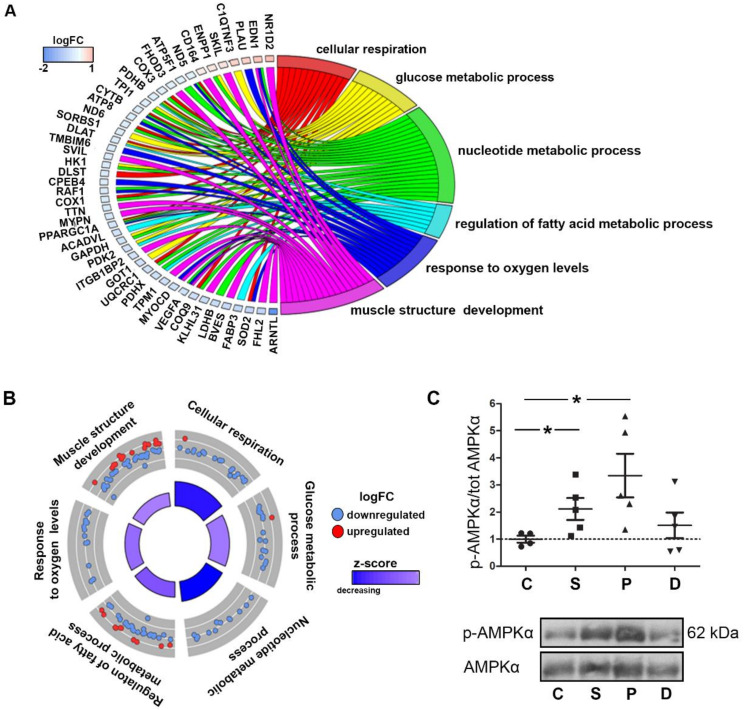
Decreased metabolism and muscle structure are enriched by serotonin, pergolide, and dexfenfluramine treatment. AMPK activation by serotonin and pergolide. (**A**): Chord plot shows 46 common DEGs and their overlap between significant Gene Ontology (GO) terms. (**B**): Circular plot is colored according to z-score and highlights logFC of DEGs among significant GO terms. The outer circle shows a scatter plot for each term of the logFC of the assigned genes. Red and blue dots display individual up-regulated and down-regulated genes, respectively. The inner rectangles are sized to positively correlate with the significance of each GO term, and colored to represent the overall direction of change in expression of each individual term. (**C**): Western blotting detection of phosphorylation of AMPKα and total AMPKα in TV leaflet total protein extracts. Data are reported as dot plots where each dot represents a rabbit and the horizontal bar indicates the mean ±SEM. * *p* < 0.05, determined with one-way ANOVA followed by Dunnett’s multiple comparison test (n = 5 for each treatment).

**Figure 4 ijms-21-04003-f004:**
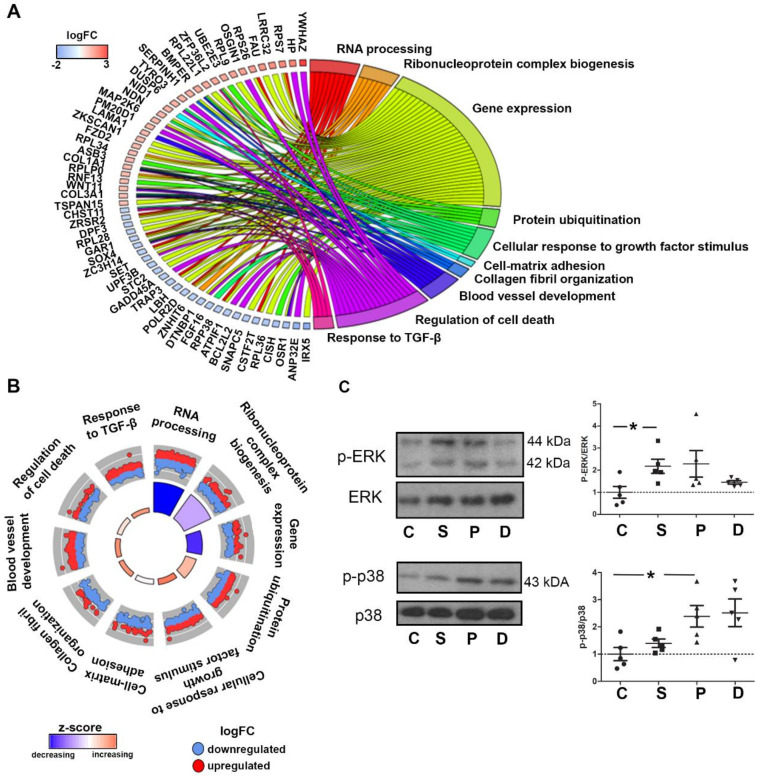
Pergolide and dexfenfluramine treatment promote a decrease in biosynthetic process and an increase of response to growth factor stimulation: (**A**): Chord plot indicates common DEGs for pergolide and dexfenfluramine with at least a 2-fold change compared to control tricuspid leaflets and related to significant GO terms. (**B**): Circular plot indicates GO terms that are likely to be decreased or increased by pergolide and dexfenfluramine. (**C**): Western blotting detection of phosphorylated ERK (p-ERK), total ERK, phosphorylated p38 (p-p38), and total p38 in TV leaflet total protein extracts. Data are reported as dot plots, where each dot represents a rabbit and the horizontal bar indicates the mean ± SEM. * *p* < 0.05, determined with one-way ANOVA followed by Dunnett’s multiple comparison test (n = 5 for each treatment).

**Figure 5 ijms-21-04003-f005:**
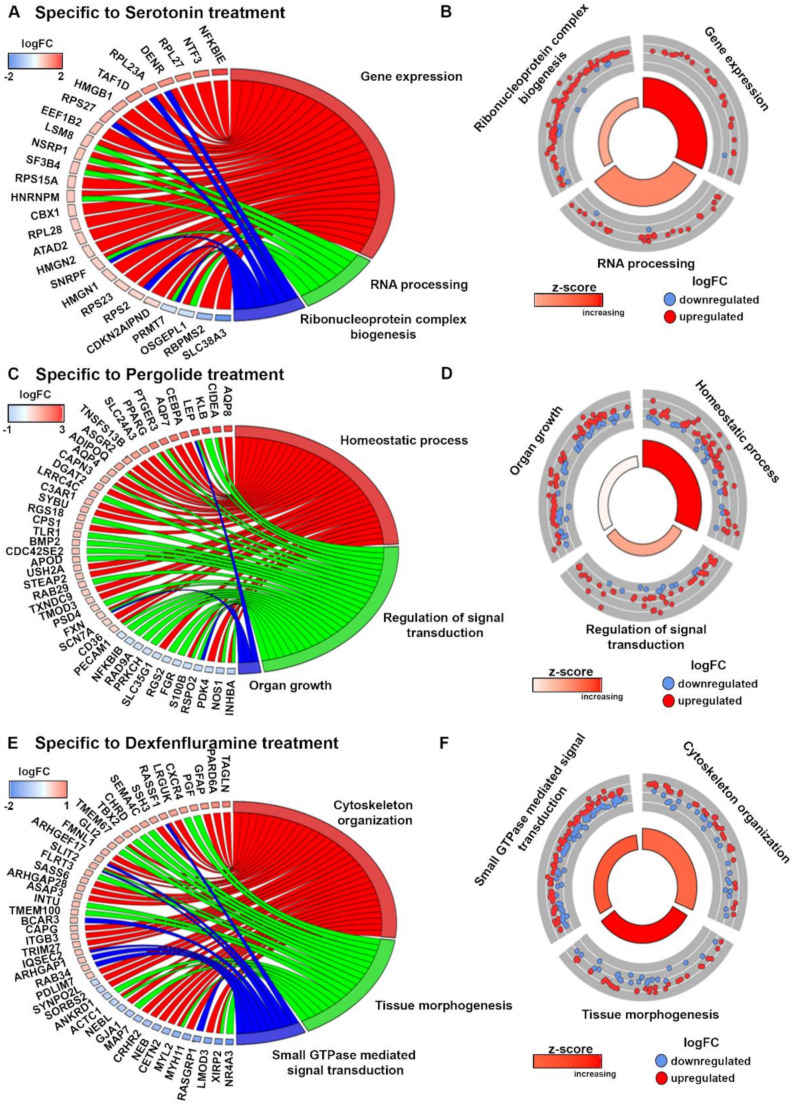
Enrichment of specific biological processes by each treatment. (**A**): Chord plot indicates specific DEGs for serotonin treatment with at least a 1.5-fold change compared to control tricuspid leaflets related to significant GO terms. (**B**): Circular plot indicates GO terms that are likely to be increased by serotonin. (**C):** Chord plot indicates specific DEGs for serotonin treatment with at least a 1.6-fold change compared to control tricuspid leaflets related to significant GO terms. (**D**): Circular plot indicates GO terms that are likely to be increased by pergolide. (**E**): Chord plot indicates specific DEGs for dexfenfluramine treatment with at least a 1.5-fold change compared to control tricuspid leaflets related to significant GO terms. (**F**): Circular plot indicate GO terms that are likely to be increased by dexfenfluramine.

**Figure 6 ijms-21-04003-f006:**
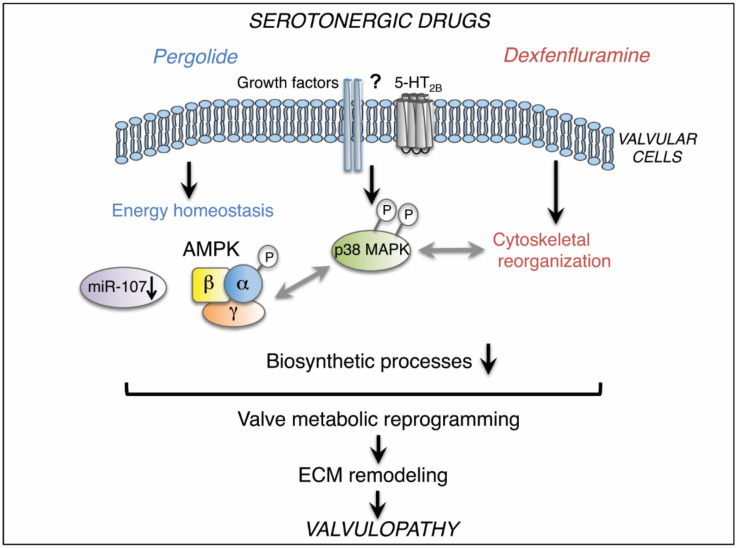
Schematic summary: chronic use of pergolide or dexfenfluramine activates p38 MAPK in valvular cells. Specifically, pergolide triggers AMPK- and miR-107-dependent homeostasis processes, whereas dexfenfluramine affects cytoskeletal organization. Drug-induced alterations of valve homeostasis and cytoskeleton are associated with net inhibition of biosynthetic processes that reflect valve metabolic reprogramming, leading to ECM remodeling and valvulopathy. Gray arrows represent hypothetical links.

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
