# Peer review of "Dexfenfluramine and Pergolide Cause Heart Valve Disease via Valve Metabolic Reprogramming and Ongoing Matrix Remodeling"

_ijms, 2020, doi:10.3390/ijms21114003_

Round 1

Reviewer 1 Report

line 74-76: Please explain how these concentrations have been selected (i.e. literature?)

line 132: Fig. 2 caption: there i s a space missing

line 131-140: The D graph is not clear. How are the 5 animals reported? Furthermore, the level of misregulation of the selected genes is below the 2-fold change threshold; how biologically significative is such value? Finally, the 'D' is lacking.

line 137: it does not seem to be a dot plot.

line 175: TGF-@.

line 182: There is no such figure. Same problem in Fig.4 caption. Please correct

Figure 4: It is very difficult to appreciate the increase in the p-p38 fraction form the picture.
Is there also a general increase in the total amount of p38 after the treatment with the 2 drugs? Did the Authors normalize the expression to some internal reference? 

line 218: in-vivo instead of in vivo (adjective).

line 315: please specify the age of the animals.

Reviewer 2 Report

Summary: The goal of the manuscript by Oury, et al., is to determine potential pathological mechanisms that underlie valvular heart disease (VHD) induced by serotonin, pergolide (an ergot derivative previously used to treat Pakinson’s disease), and dexfenfluramine (a withdrawn anorectic). These compounds are reported to have high affinity for 5HT-2B serotonin receptors, a hypothesized mediator of VHD, and are used here to induce VHD in rabbits after 16 weeks of treatment. Representative changes in the pathology of aortic, mitral and tricuspid valves are first presented. The rest of the manuscript focuses on common and unique transcriptional changes in the tricuspid valve in response to the three compounds and show serotonin and pergolide induce AMPK phosphorylation while pergolide uniquely leads to increased p38 phosphorylation.

Broad comments: This paper attempts to addresses an important problem in the field and is the first to examine transcriptional changes in VHD-causing compounds that potentially overlap by binding to and activating 5HT-2B receptors. Such a contribution would be important in understanding VHD pathogenesis. It is well written and easy to understand. While the pathological changes presented are visually profound and the transcriptional profiles between the different treatments are quite different, there are a few technical and statistical issues that do not support many of the authors’ interpretations and conclusions. In addition, the study is correlative in nature and could be strengthened testing the hypothesis of whether 5HT-2B receptors contribute to the drug-induced transcriptional changes.

Specific comments:

  1. It is surprising that transcriptional analysis was performed on the tricuspid valve given no significant pathological change is demonstrated for TV leaflet area (Fig. 1X) other than observational collagen remodeling. Are there any significant and changes in valve pathology by the three groups (other than AoV leaflet area (Fig. 1V)), and specifically for the tricuspid valve? It is necessary to show significant and quantifiable pathological changes are occurring in the TV to convince the reader that any transcriptional differences found are pathologically relevant. This group’s 2015 Int. J. Cardiol. paper (reference 5) demonstrates serotonin significantly increases aortic, mitral, and tricuspid valve thickness but no increase in valve areas are induced by serotonin in rabbits used in this experiment. This part of the study may also suffer from being underpowered (n=3 animals).
  2. Relativistic terms discussing pathological changes should be carefully re-evaluated and supported by statistical evidence. For example, statements such as: accumulation of GAG-rich deposits was “comparable” (lines 84-85); “high levels” of CTGF induced by pergolide and dexfenfluramine but not serotonin (lines 87-89); and the summary statement that “pergolide and dexfenfluramine caused distinct valvular lesions that are also histologically different from those induced by serotonin” (lines 91-92) are observational as presented and could be considered biased without proper quantification.
  3. Is the presence of metaplasia nodules in the aortic hinge for the three treatments (Fig 1U) significant? If so, what is the rationale for not investigating transcriptional changes in the aortic hinge? Similarly, why not study the AoV found significant for pergolide, which may be informative for pergolide-specific changes?
  4. RNA-seq analysis may be compromised due to low RNA quality. RNA integrity numbers for samples are 5.8-7.6 (line 356), lower than a RIN value of at least 8.0 typically recommended for RNA-seq. Please discuss how these RIN values may affect transcriptomic data and analysis. Did specific control or treatment groups display higher/lower RIN values compared to other treatment groups, and if so, how might that explain the resulting enriched gene ontology terms?
  5. The supplemental table S1 listing all DEGs (line 121) was not provided. Table S1 is instead a list of qPCR primers (described as Table S2, line 392).
  6. Although RNA-seq data was confirmed to be deposited into GEO (GSE113082, line 380), the data is private and it appears a review token was not provided to examine the data further.
  7. RNA-seq data would provide important gene-gene comparisons that are relevant for the overall hypothesis and proposed mechanisms. What are the expression levels for the different serotonin receptors in the tricuspid valve compared to their known relative affinities for the drugs used? What are the levels of Tgfb1 and TGF-beta related signaling molecules? Downstream p38 gene targets? miR107 targets?
  8. Discussion of previous valve disease transcriptomics results are not discussed. How do the RNA-seq results here compare with published data? (e.g. PMID: 27495158, 25547111)
  9. The authors struggle throughout with whether or not pergolide or dexfenfluramine act on 5HT (and more specifically 5HT-2B) receptors in differentially regulating transcriptional output compared with serotonin. The functional significance of 5HT-2B receptors in VHD is a central problem for the field. This study could be strengthened considerably if all or part of the transcriptional responses from pergolide or dexfenfluramine treatment could be reversed with a serotonin receptor antagonist or similar approach. Otherwise, the authors are left to speculate that differences may be due to affinity differences for 5HT receptors, or, are more likely due to indirect, non-5HT, or toxic effects.
  10. Similarly, the significance of the study could be much improved by showing the proposed mechanistic pathways (AMPK, p38, miR107) are causal through targeted inhibition experiments as proposed in the discussion.
  11. How do the authors reconcile that gene ontology enrichment for RNA processing, Gene expression, and Ribonucleoprotein complex biogenesis are upregulated by serotonin but repressed by pergolide and dexfenfluramine?
  12. Please include numerical z-score ranges for or all RNA-seq plot figures (Figs 3B, 4B, 5B,D,F). Such values are important in properly evaluating the data.
  13. What do the concentric gray circles represent in the RNA-seq circle plots (Figs 3B, 4B, 5B,D,F)?
  14. In lines 248-249, the number of DEGs from pergolide or dexfenfluramine that are shared with serotonin DEGs should be ~5-6% (169/2692 and 169/3020 DEGs), not 30% as written.
  15. What is the rationale for using only female rabbits?

Round 2

Reviewer 2 Report

The authors have adequately addressed my concerns.